# A systematic review of post COVID-19 condition in children and adolescents: Gap in evidence from low-and -middle-income countries and the impact of SARS-COV-2 variants

**Nina Dwi Putri**[1,2,3]*, **Ida Safitri Laksanawati**[4,5], **Dominicus Husada**[6,7],
**Nastiti Kaswandani**[1,2], **Ari Prayitno**[1,2], **Rina Triasih**[4,5], **Irma Sri Hidayati**[4,5], **Retno Asih**[6,7],
**Robby Nurhariansyah**[7], **Fabiola Cathleen**[2], **Dwiana Ocviyanti**[8,9], **Sri Rezeki Hadinegoro**[1,2],
**Dan Pelicci**[3], **Julie Bines**[3], **Stephen M. Graham**[3]

1 Department of Pediatrics, Cipto Mangunkusumo Hospital, Jakarta, Indonesia, 2 Department of Pediatrics, Faculty of Medicine, Universitas Indonesia, Depok, Indonesia, 3 Department of Paediatrics, University of Melbourne and Murdoch Children's Research Institute, Royal Children's Hospital, Melbourne, Victoria, Australia, 4 Department of Pediatrics, Dr. Sardjito Hospital, Yogyakarta, Indonesia, 5 Department of Pediatrics, Faculty of Medicine, Nursing and Public Health, Universitas Gadjah Mada, Yogyakarta, Indonesia, 6 Department of Pediatrics, Dr. Soetomo Hospital, Surabaya, Indonesia, 7 Department of Pediatrics, Faculty of Medicince, Universitas Airlangga, Surabaya, Indonesia, 8 Department of Obstetrics and Gynaecology, Cipto Mangunkusumo Hospital, Jakarta, Indonesia, 9 Department of Obstetrics and Gynaecology, Faculty of Medicine, Universitas Indonesia, Jakarta, Indonesia

* ninadwip@gmail.com

## Abstract

The long-term health consequences following COVID-19 have largely been reported in adult populations living in high-income countries. We therefore did a systematic review of post COVID-19 condition symptoms reported in children and adolescents (<18 years), aiming to identify and include publications from low- or middle-income countries (LMICs). From EMBASE, Medline, and Pubmed until the 30th of October 2023, we searched all studies reporting original and complete data of long-term outcomes of at least 20 children or adolescents under 18 years of age with a history of confirmed acute COVID-19 infection. We excluded non-English publications, pre-prints, unreviewed articles, grey literature, studies with inaccessible full text, and those limited to a specific population. Risk of Bias was assessed using STROBE guidelines for observational studies. We used descriptive narrative analysis to summarize the findings. Forty studies reporting 825,849 children and adolescents; the median age of those with persistent symptoms was consistently in the adolescent age range but not all studies included young children (<5 years). Only one study, with 58 participants aged 6-17 years, population was from a LMIC. Studies relied on symptom reporting rather than objective measures of organ dysfunction. The definition of post COVID-19 condition varied; most studies used persistent symptom duration of two or three months or more. However, since the symptom onset was not specified, it was difficult to identify which study is truly consistent with WHO's definition of post COVID-19 condition. Prevalence of post COVID-19 condition ranged from 1.8% to 70% but with marked

**Data availability statement:** All relevant data are within the manuscript and its Supporting Information files.

**Funding:** This research received funding from the Indonesia Endowment Fund for Education Agency on the Partnership in Research Indonesia and Melbourne (PRIME funding code PRJ-120/LPDP/2021) through DO. The funders had no role in study design, data collection and analysis, publication decisions, or manuscript preparation.

**Competing interests:** The authors have declared that no competing interests exist.

heterogeneity between study populations and reporting criteria including the severity of acute COVID presentation. Most studies were undertaken when the Alpha variant was the predominant strain. The prevalence of post COVID-19 condition ranged from 6.7% to 70% in the Alpha variant-, 23% to 61.9% in the Delta-, 17% to 34.6% in the Omicron-, and 3.7% to 34% in the Other-variant predominated studies. The most reported symptoms were fatigue (70%), headache (37.5%) and respiratory symptoms (35%); fatigue was most reported in all variant subgroups. Only half of the studies included a control group. The variations in study population, reporting methods, reliance on symptom reporting alone and lack of control groups make it challenging to determine the impact of COVID-19 on post COVID health in children and adolescents. The lack of data from LMIC populations especially infants and young children is a major gap.

## Introduction

Globally, one-fifth of Coronavirus disease 2019 (COVID-19) reported cases have been in children and adolescents (<18 years) [1]. COVID-19 in children is usually less severe and associated with a lower mortality than in adults [2]. However, severe disease and COVID-related mortality have been observed in neonates and infants, children with comorbidities and those living in low- or middle-income countries (LMICs) [3–5]. Furthermore, there is concern regarding long-term health consequences following acute COVID-19 infection in children and adolescents based on observational evidence in adults with COVID-19 as well as previous experience of severe acute respiratory syndrome (SARS) and Middle East Respiratory Syndrome (MERS) [6,7].

Post COVID-19 condition in adults have been receiving increasing attention, with over 100 symptoms documented [8]. A meta-analysis of 22 studies found that prolonged fatigue, joint pain, anosmia, headache, and myalgia, were recorded in 21.2%, 15.4%, 9.7%, 8.9%, and 5.6%, of adult COVID-19 survivors, respectively [9]. Similarly, a 2024 meta-analysis identified post COVID-19 symptoms in 30% of 7,912 adult participants, with fatigue being the most prevalent symptom, affecting 28.0% of individuals two years following the acute infection. However, the study reported high heterogeneity and limits generalizability [8]. It is important to have similar analysis for children and to contrast with data in adults.

By the end of 2023, there were eight published systematic reviews of post COVID-19 condition in children [10–17]. The most recent systematic review included over 15,000 study participants and reported that over 16.2% of children and adolescents experienced post COVID-19 condition symptoms [15]. Almost all reported study populations live in high-income countries (HICs) and the definitions of post COVID-19 condition used by studies were highly variable. A standardized clinical case definition was published by the World Health Organization (WHO) in 2023 [18] but this has not been applied in previously published reviews.

There are other variables between populations related to COVID epidemiology that may impact on post COVID-19 conditions such as changes over time of the effect of different SARS-COV-2 variants and major differences in the policy and application of COVID-related restrictions, including differences between and within HICs and LMICs. One important example is the wide variation in school closure and re-opening [19–23]. The lack of evidence from LMICs was noted as a concern since the WHO's webinar and expert meetings in August 2022 [24]. While there is a high burden of respiratory infections due to a wide range of pathogens in children living in LMICs, studies of long-term symptoms are rare [25,26]. Furthermore, most studies to date have focused on describing the symptoms and less on the

management of post COVID-19 condition [23]. There is often a lack of access to techniques that measure organ dysfunction at a more sophisticated level beyond reporting of symptoms, especially in LMICs [23].

A prospective cohort study is underway in three sites in Indonesia that is comprehensively assessing abnormalities in a range of bodily systems and quality of life in children and adolescents at follow-up up to 12 months post-COVID [27]. To identify current knowledge gaps and inform future research directions, we conducted a systematic review that aimed to provide an update of the current evidence on post COVID-19 condition in children and adolescents, with consideration of the Gross National Income (GNI) of the country represented by the study population, the dominant variant of SARS-COV-2 at the time of the study and application of the recent WHO definition.

## Methods

This systematic review was reported using Preferred Reporting Items for Systematic Reviews and Meta-analyses (PRISMA) for study design, search procedure, screening, and data reporting guidelines [28].

### Eligibility criteria

**Types of studies.** We included all literature that reported original and complete data of at least 20 children or adolescents, regardless of study design (with or without a control group) or setting. As a result, systematic reviews, reviews, case series, and case reports were excluded.

**Types of participants.** Children and adolescents less than 18 years of age with confirmed acute COVID-19 infection (hospitalized and non-hospitalized) as determined through either positive SARS-COV-2 Reverse-transcriptase Polymerase Chain Reaction (RT-PCR), antigen test, serology test, or clinical diagnosis by physician. An age cut-off of 18 years was chosen as some search filters still use <18 years old to categorize children and adolescents.

**Types of outcome measures.** We focused on the long-term outcomes of children and adolescents previously infected with SARS-CoV-2. Any symptoms and definitions for post COVID-19 condition was accepted; the WHO post COVID-19 condition expert consensus definition in children and adolescents was only published recently, in February 2023 [18].

**Exclusion criteria.** We excluded publications in a language besides English, pre-prints/pre-published, unreviewed articles, grey literature, unattained full-text, and publications that only provided results to a specific population such as patients with immunodeficiency and Multisystem Inflammatory Syndrome in Children (MIS-C) or reported specific COVID-related symptoms or specific organ involvement only (e.g., radiography or laboratory results, immunology profiles, psychiatric problems only). We also excluded grey literature defined as materials produced outside academic publishing such as government documents, reports, thesis and dissertations, or conference papers.

### Search strategy for studies identification

**Electronic searches.** A single reviewer (NDP) conducted a systematic search with guidance from a medical librarian using the Medline (via Ovid), Embase, and PubMed databases from pandemic onset until the 30 October 2023 (S1 File). The search strategy was structured using Medical Subject Heading (MeSH) terms including "COVID-19", "children under five years old", "overall category of children age", "adolescents", "post COVID-19 condition", "long-term outcome", and "post-COVID-19". All keywords were combined using the Boolean logic operation "OR"/ "AND". Advanced search terms included "tw" (text word), "kf" (keyword heading word), and "hw" (heading word) for Medline (from Ovid), and "tw"

(text word), "kf", "hw", and "dq" (candidate term word) for Embase. Additional records were identified through a bibliography search of available systematic reviews and similar article recommendations up to February 13, 2024.

**Selection of studies.** Study selection was done by lead author (NDP) and reviewed by a second author (SMG or JEB). We used Endnote X9 desktop to combine publications from all databases and remove duplicates. Studies remaining after duplicate removal were screened for eligibility based on their titles and abstracts using Endnote X9 desktop. Publications deemed irrelevant to the topic or objectives of this review were excluded. The process then proceeded to the assessment of full-text articles based on inclusion and exclusion criteria. Additional reference searching was done manually from included studies' bibliographies and similar article suggestions.

## Data collection and analysis

**Data extraction.** One investigator (NDP) extracted data, which was subsequently verified by other investigators (SMG and JEB) for quality control. The characteristics of the included studies were manually extracted into a Microsoft Excel table, including study details (author, time of data collection, study location, study design), participant details (number of subjects, sex, age of overall participants, age of participants in post COVID-19 condition group and control if any, acute SARS-CoV-2 confirmation method, the severity of acute COVID-19,), and outcome details (post COVID-19 condition definition, duration of follow up, post COVID-19 symptom prevalence, and the three most frequently reported persistent symptoms with prevalence of each. Data and numbers of control populations were extracted if available for comparison with cases including age, sex and proportions with persistent symptoms.

**Missing data.** When data on essential outcomes were reported as unclear or missing, we requested additional data to the included studies' corresponding authors. If no response was received, the study's inclusion was rediscussed based on its relevance to the overall analysis.

**Assessment of risk of bias in included studies.** Quality assessments within studies were done after the data extraction of all included studies by one investigator (NDP) in consultation with second reviewer (SMG or JEB). The quality of the study was assessed based on the study adherence using Strengthening the Reporting of Observational Studies in Epidemiology (STROBE) recommendations [29]. To aid the assessment, we quantified "Yes" as two, "Partial" as one, and "No" as zero. The scoring results were considered in determining the studies' quality, and categorized as follows: low (score ≤ 29), moderate (score 30–39), and high quality (score ≥ 40). However, this scoring was not be the sole determinant; subjective assessments were integrated into the final evaluation of each study's overall quality.

**Data synthesis.** We used a descriptive narrative approach to analyze the persistent symptoms reported in all included studies, based on the WHO clinical case definition of post COVID-19 condition in children and adolescents. The definition specifies a symptom lasting at least two months with onset within three months of acute COVID-19 presentation in individuals with a history of confirmed or probable SARS-CoV-2 infection [18]. Symptoms may include new onset following initial recovery, persistence, fluctuation, or relapse from the initial illness. Reported symptoms were categorized into fourteen entities:

1. Fatigue or tiredness

2. Respiratory symptoms: including dyspnea or shortness of breath

3. Headache

4. Weakness or asthenia and exercise intolerance or reduced physical resilience

5. Musculoskeletal symptoms: including myalgia or joint pain

6. Loss of appetite

7. Neurocognitive problems: including difficulty in concentration or memory

8. Neuropsychiatric problems: including sleeping difficulties, depression, or anxiety

9. Sensory problems: including loss of smell or loss of taste

10. Systemic problems: including fever, chills, syncope, dizziness, or weight loss

11. Other respiratory symptoms: including cough, rhinitis, or nasal congestion

12. Gastrointestinal (GI) or esophageal problems

13. Dermatological problems

14. Cardiovascular problems

We summarized the overall prevalence and types of post COVID-19 conditions. All tabulations were visualized with tables and graphs when feasible. We used the World Bank's GNI stratification to categorize the economies of the countries of the study population resided and data were collected. Countries with GNI per capita from $1,136 to $4,255 were regarded as LMICs, $4,256 to $13,845 as upper-middle-income countries (UMICs), and $4,256 or more as HICs [30]. We utilized the Our World in Data website to match the variants of SARS-CoV-2 with each study location and period [31]. If a study encompassed multiple SARS-CoV-2 variants, we analyzed the study according to the predominating Variant of Concern (VOC) density it covers. If the SARS-COV-2 variants were not available on the website, we reported them as "not identifiable" (referring to the first and second waves of the pandemic).

We did not conduct a meta-analysis due to the wide heterogeneity in study design (such as varying definitions of post COVID-19 condition, timing and method of follow-up assessments) and in study participants (including variations in reporting by age groups and the severity of acute COVID).

## Institutional review board

This study did not require ethical clearance.

## Results

### Overall findings

Fig 1 presents the findings of the search strategy as a PRISMA flow diagram. From 951 publications identified, we removed 185 duplicates and 521 ineligible publications using Endnote X9 automation tools. Of 245 abstracts and titles screened, we retrieved 50 publications for eligibility assessment of which we excluded 28 that were reviews or systematic reviews (n = 17), studies which only explored specific symptoms or organ involvement (n = 8), studies which only reported broad symptoms in the text. We identified an additional eighteen studies that met inclusion criteria from reference searching of reviews. Therefore, there were 40 publications were included in the final analysis [32–71] of which six studies were published in 2023 [30,31,34,58,61,63]. Table 1 lists the main study characteristics and findings by publication including age and sex characteristics in controls when reported.

Of the 40 studies, 31 (77.5%) were cohort studies, including 22 (55%) prospective [35,39,41,43,47–50,54–61,64,65,68–71], eight (20%) retrospective [33,36,37,40,53,63,66,67] and one (2.5%) ambidirectional cohort study [62]. The remaining nine (22.5%) were

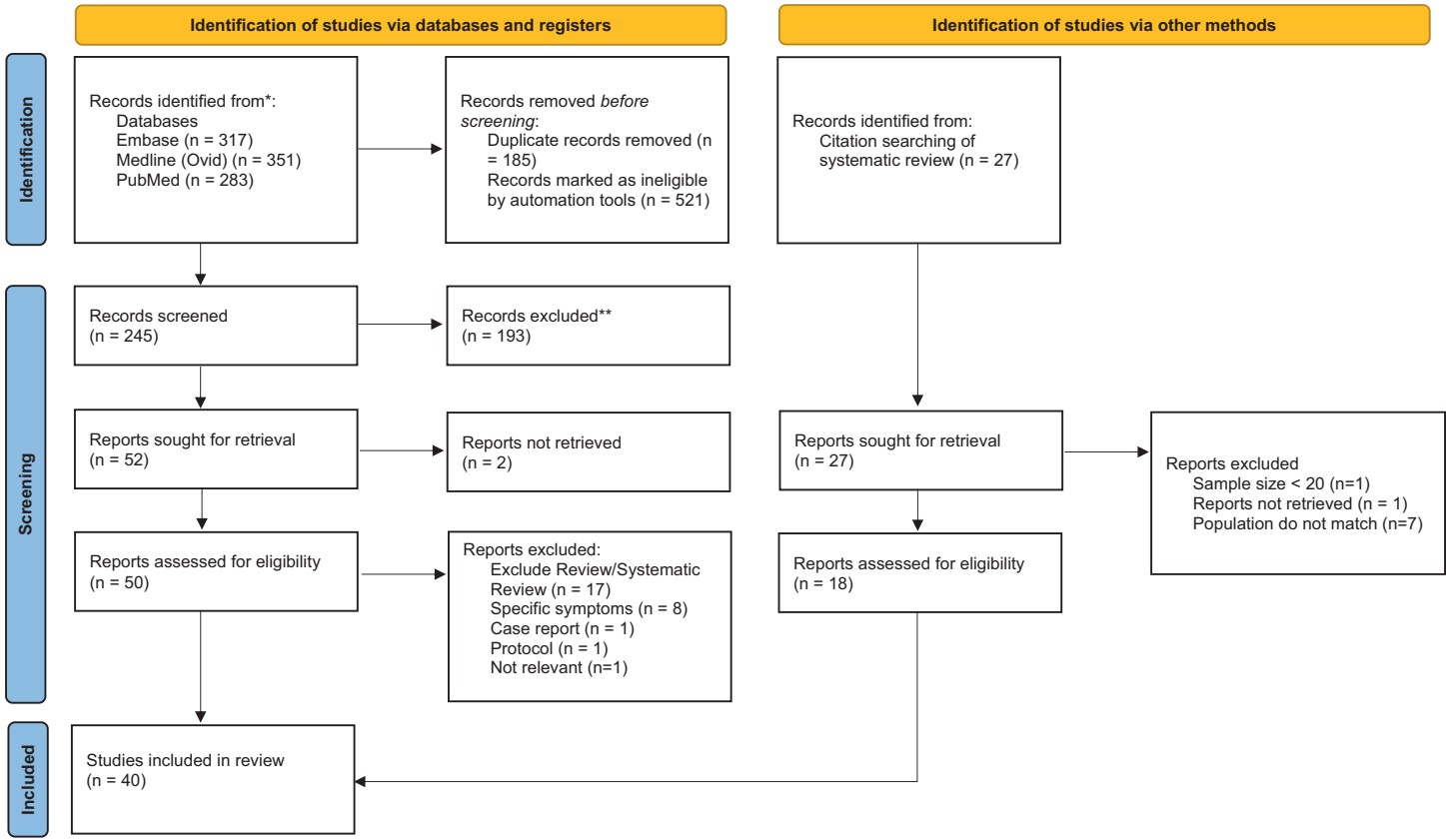

**Fig 1. The PRISMA flow diagram of the literature search.**

cross-sectional studies [32,34,38,42,44–46,51,52]. Around half (21, 52.5%) of studies included a control group [32,33,37,39,40,43,46–48,50–53,56,57,60–62,65,68,71]; cases and controls had similar proportions of females and age distribution similar in all but two studies [37,62].

In total, 825,849 children and adolescents with a history of acute COVID-19 infection were included in this study. The median sample size of COVID-19-infected participants was 181, ranging from 16 to 781,419 [53,66]. Eleven studies included less than 100 participants [34,35,39,41,42,49,52,63,66,67,69]. The proportion of females in these studies ranged from 39% to 66% [49,67]. The age distribution ranged from 11 days in one study [41] up to 18 years in most studies, skewed towards older children and adolescents. Ten studies did not include infants or young children (<5 years) and four of these were limited to adolescents (10-18 years); therefore, median or mean age of study participants was commonly within the adolescent age range (18/31 studies, 58.1%). Only six studies documented the age range of those identified with a post COVID-19 condition and the range was similar to the overall study population [34,35,37,38,51,52].

All except two studies include PCR as one of the SARS-CoV-2 confirmatory tests in acute conditions; one study compared seropositive to seronegative patients [61], and the other did not state how they define acute infection [63]. Five studies included only hospitalized patients [34,35,41,47,69], ten on outpatients [32,33,38–40,42,50,60,63,70], and 21 studies included both hospitalized and non-hospitalized patients [36,37,43–46,48,49,51–55,57–59,62,64–67], while the remaining four did not specify [56,61,68,71]. The prevalence of post COVID-19 symptoms ranged from 15% to 45% among studies in only hospitalized patients [32,39] and from 13% to

**Table 1. Characteristics of 40 studies included in the analysis.**

| No | Study | Country income status | SARS-COV-2 confirmation | Study design | N | Female % | Age of participants | Persistent symptoms % | Duration of follow up | Three symptoms most reported % | Inpatient/ Outpatient/ Both |
|---|---|---|---|---|---|---|---|---|---|---|---|
| 1. | Adler et al [32] | HIC | PCR (+) or antigen test (+) | CS Case Control | 3240 1148 2092 | 62.1% 63.7% 61.3% | Range: 5–18 years Mean 10.8 years Mean: 9.5 years | 43.7% 33.3% | Mean: 4.39 ± 1.5 months | Headache (18.4%) Weakness (15.1%) Fatigue (12.3%) | Outpatient |
| 2. | Ahn et al [33] | HIC | PCR (+) | RC Case Control | 211 106 105 | 40.6 41.9 | Range: 6 months–18 years Median: 3 years Median: 2 years | 17% 4.8% | >12 weeks | Abdominal pain (6.6%) Mild fever (4.7%) Respiratory symptom (4.7%) | Outpatient |
| 3. | Asadi-Pooya, et al [34] | LMIC | Symptomatic and PCR (+) | CS | 58 | 58% | Range: 6–17 years Mean (+SD): 12.3 (3.3) | 44·8% | Min 3 months after hospital discharged | Fatigue (20.7%) Shortness of breath (12%) Exercise intolerance (12%) | Inpatient |
| 4. | Ashkenazi-Hoffnung, et al [35] | HIC | PCR (+) or antibody (+) | PC | 90 | 42% | Range: 0– ≤18 years Mean (+SD): 12 (5) years | N/A | Median of 112 days (range: 33–410 days) | Fatigue (71.1%) Dyspnea (50.0%) Myalgia (46%) | Inpatient |
| 5. | Baptista de Lima et al [36] | HIC | PCR (+) | RC | 144 | 45.6% | Range: 0–18 years Mean: 79 months | 13.3% | >4–24 weeks | Fatigue (7.5%) Behavioral changes (4.5%) Sleep disturbance (3.7%) (12 weeks) | Both |
| 6. | Bergia et al. [37] | HIC | PCR (+) or antigen (+) or antibody (+) | RC Case Control | 549 451 98 | 45% 43% | Range 0–18 years Median: 4.0 years Median 7.8 years | 14.6% 19.4% | Median 351 days | Fever (80.3%) Cough (59.1%%) Asthenia (73.3%) | Both |
| 7. | Bloise et al. [38] | HIC | PCR (+) | CS | 1413 | 62% | Range:0–18 years Median (IQR): 10 (6–13) years | 20% | Mean 87.49 ± 56.44 days after diagnosis | Asthenia (39.9%) Difficulty in concentration and memory (21.3%) Trouble sleeping-depression and other neuropsychiatric disorders (17.8%) | Outpatient |
| 8. | Blomberg et al [39] | HIC | PCR (+) and Antibody (+) | PC Case Control | 33 16 17 | 56% N/A | Range 0–15 years Median (IQR): 8 (6–12) | 13% N/A | 6 months | Disturbed taste or smell (13%) Stomach upset (6%) | Outpatient |
| 9. | Borch et al. [40] | HIC | PCR (+) | RC Case Control | 30,121 15,041 15,080 | N/A | Range: 0–17 years Mean ages similar | 25.3% 22.8% | Minimum 4-weeks from PCR test | Fatigue (11%) Loss of smell and taste (10%) Headache (7%) | Outpatient |
| 10 | Bossley et al [41] | HIC | PCR (+) | PC | 71 | 41% | Range: 11 days– 17 years Mean: 6.7 years | 15% | >4 weeks | Dry cough (7%) Shortness of breath (6%) Fatigue (4%) | Inpatient |
| 11. | Brackel et al. [42] | HIC | PCR (+) or antibody (+) or clinical diagnosis or unknown | CS | 89 | N/A | Range: 2–18 years Median: 13 years | N/A | > 12 weeks | Fatigue (86.5%) Dyspnoea (55.0%) Concentration difficulties (44.9%) | Outpatient |
| 12. | Buonsenso et al (a) [43] | HIC | PCR (+) | PC Case Control | 286 249 37 | 49% 53% | Range:0–18 years Mean: 10.4 years Mean: 10.5 years | 32% N/A | Median: 77 days post diagnosis | Insomnia (19%) Asthenia (14%) Cough (12%) (6–9 months post diagnosis) | Both |
| 13. | Buonsenso, et al (b) [44] | HIC | Clinical diagnosis or antibody (+) or PCR (+) or suspected | CS | 510 | 56% | Range: 1–18 years Mean: 10.3 years | 25.3% | Mean 8.2 months | Tiredness and weakness (87%) Fatigue (80%) Headache (79%) | Both |

*(Continued)*

**Table 1.** (Continued)

| No | Study | Country income status | SARS-COV-2 confirmation | Study design | N | Female % | Age of participants | Persistent symptoms % | Duration of follow up | Three symptoms most reported % | Inpatient/ Outpatient/ Both |
|---|---|---|---|---|---|---|---|---|---|---|---|
| 14. | Buonsensoc et al (c) [45] | HIC | PCR (+) | CS | 129 | 48.1% | Range: 0–18 years Mean: 11 ± 4.4 years | 66.7% | >60 days after diagnosis, average: 162.5 ± 113.7 days | Insomnia (23.3%) Headache (23.3%) Weight loss (16.7%) | Both |
| 15. | Erol et al [46] | UMIC | PCR (+) or antibody (+) | CS Case Control | 216 121 95 | 46.3% 47.4% | Range: 0–18 years Median: 9.16 years Median: 8.42 years | 37.2% N/A | >1 month | Chest and backache (51.1%) Dizziness ± syncope (15.6%) Palpitation (11.1%) | Both |
| 16. | Fink et al [47] | UMIC | PCR (+) or antibody (+) | PC Case Control | 105 53 52 | 58% 60% | Range:8–18 years Median: 14.7 years Median: 14.8 years | 23% N/A | Median: 4.4 months (0.8 – 10.7) months post diagnosis | Headache (18.9%) Tiredness (9.4%) Dyspnoea (7.5%) | Inpatient |
| 17. | Funk et al [48] | UMIC and HIC | PCR (+) | PC Case Control | 3585 1884 1701 | 47% NR NR | Range:0–18 years Median: 3 years 73% < 10 years 79% < 10 years | 5.8% | 90–120 days after SARS-CoV-2 testing | Fatigue (1.1%) Cough (0.7%) Dyspnea (0.7%) | Both |
| 18. | Gonzales et al [49] | HIC | PCR (+) or antibody (+) | PC | 50 | 66% | Range: 5.5–17.9 years Median: 14.1 years | N/A | >12 weeks after infection | Fatigue (100%) Neurocognitive disorder (74%) Muscular weakness (74%) | Both |
| 19. | Haddad et al [50] | HIC | PCR (+) and/or Symptomatic with sero-conversion | PC Case Control | 544 210 infected 334 exposed | 47% 52% | Range: 1–18 years 72% < 14 years 76% < 14 years | 21.4% 8.4% | 11-12 months after infections | Fatigue (3.3% Reduced physical resilience (2.9%) Breathlessness (1.9%) | Outpatient |
| 20. | Kikkenborg Berg, et al [51] | HIC | PCR (+) | CS Case Control | 28,270 6,630 21,640 | 58% 57% | Range: 15–18 years Median: 17.6 years Median: 17.5 years | 61.9% 57% | At least 2 months after infection | Fatigue (13%) Loss of appetite (6.9%) Headache (6.5%) | Both |
| 21. | Knoke et al [52] | HIC | PCR (+) and/or antibody (+) | CS Control for baseline characteristics only | 70 | N/A | Range: 5–18 years Mean (+SD): 10.8 (3.3) | 27.1% | 2.6 months average | Breathing problems (5.7%) Fatigue (7.1%) Loss of smell/taste (5.7%) | Both |
| 22. | Kompaniyets, et al [53] | HIC | PCR (+) or an ICD-10-CM code of B97.29 or U07.1 | RC Case Control | 3,125,676 781,419 2,344,257 | 50% 50% | Range: 0–17 years Median: 12 years Median: 12 years | N/A | 60 days – 365 days | Respiratory signs and symptoms (10.9%) Musculoskeletal symptoms (8.7%) Gastrointestinal and esophageal disorders (3.9%) | Both |
| 23. | Kuczborska et al [54] | HIC | PCR (+) or antigen (+) | PC | 147 | 47% | Range: 4 months–17 years Median 9 years | 34.6% among immuno-competent | Min 3 months | Fever (61%) Cough/Rhinitis (45.5%) Fatigue (42.9%) | Both |
| 24. | Matteudi et al [55] | HIC | PCR (+) | PC | 137 | N/A | Range: 17 days–15 years Median: 9.3 years | 16.8% | 10–13 months after diagnosis | Asthenia (9.5%) Learning difficulties (8%) Headache (6%) | Both |

*(Continued)*

**Table 1.** (Continued)

| No | Study | Country income status | SARS-COV-2 confirmation | Study design | N | Female % | Age of participants | Persistent symptoms % | Duration of follow up | Three symptoms most reported % | Inpatient/ Outpatient/ Both |
|----|-------|----------|------------|--------|---|--------|--------|----------|----------|-------------|----------|
| 25. | Miller et al [56] | HIC | PCR (+) or antibody (+) | PC Case Control | 5032 1062 3970 | 41% NR NR | Range: 0–17 years NR NR | 4.1% 2.2% | ≥28 days after symptom onset | General symptom (30.2%) Respiratory symptoms (14%) ENT symptoms (14%) | N/A |
| 26. | Molteni et al [57] | HIC | PCR (+) | PC Case Control | 3,468 1,734 1,734 | 50% 50% | Range: 5–17 years Mean: 13 years Mean: 13 years | 1.8% N/A | 56 days | Anosmia (84%) Headache (80%) Sore throat (80%) | Both |
| 27. | Osmanov et al [58] | UMIC | PCR (+) | PC | 518 | 52% | Range:0–18 years Median: 10.4 years | 24.3% | Median: 256 days | Fatigue (10.7%) Sleep disturbance (6.9%) Sensory problems (5.6%) | Both |
| 28. | Pazukhina, et al [59] | UMIC | PCR (+) | PC | 360 | 52% | Range:0–18 years Median: 9.5 years | 20% | >6 months Median: 255 days | Fatigue (9%) Dermatological (5%) Neurocognitive (4%) (6-month follow-up group) | Both |
| 29. | Pinto Pereira et al [60] | HIC | PCR (+) | PC Case Control | 12949 6407 6542 | 62.9% 62.3% | Range: 11–17 years | 60.9% 43.2% | >6 months | Tiredness (38.4%) Shortness of breath (22.8%) Headache (18.3%) | Outpatient |
| 30. | Radtke et al [61] | HIC | Antibody (+) | PC Case Control | 1355 109 1246 | 53% 54% | Range: 6–16 years Median: 11 years 61% cases 6-11 years 56% controls 6-11 years | 3.7% 2.2% | 6 months | Tiredness (2.8%) Difficulty concentrating (1.8%) Increased need for sleep (1.8%) | N/A |
| 31. | Roge et al [62] | HIC | PCR (+) or seroconversion | Ambidirectional cohort Case Control | 378 236 142 | 45% 47% | Range: 1 month–18 years Median: 10 years Median: 2 years | 70% 24.8% | Median: 73.5 days Median: 69 days | Irritability (27.6%) Impaired attention (19.2%) Fatigue (19.2%) | Both |
| 32. | Sakurada et al [63] | HIC | N/A | RC | 54 | 57.4% | Range: 11–18 years Mean: 15.3 | N/A | Min 4 weeks | Fatigue (55.6%) Headache (35.2%) Dysosmia (30%) | Outpatient |
| 33. | Say et al [64] | HIC | PCR (+) | PC | 151 | 42% | Range:0–18 years Median: 2 years Mean: 3.7 years | 8% | 3–6 months | Post-viral cough (4%) Fatigue (2%) | Both |
| 34. | Seery et al [65] | UMIC | PCR (+) | PC Case Control | 1216 639 577 | 47% 48% | Range: 1–17 years Median: 7 years Median: 8 years | 34% 13% | >3 months | Headache (15.2%) Cough (11.1%) Fatigue (8.7%) | Both |
| 35. | Smane et al (a) [66] | HIC | PCR (+) | RC | 30 | 43% | Range: 3 months–17 years Mean: 9.2 years | 30% | Mean: 101 days | Prolonged low-grade fever (6.7%) Joint pain (3.3%) Headache (3.3%) | Both |
| 36. | Smane et al (b) [67] | HIC | PCR (+) | RC | 92 | 39% | Range: 1–18 years Median: 12 years | 51% | 1-3 months | Tiredness (38%) Loss of taste/smell (16%) Headaches (15%) | Both |
| 37. | Stephenson et al [68] | HIC | PCR (+) | PC Case Control | 6,804 3,065 3,739 | 63.2% 63.5%% 62.9% | Range: 11–17 years 41% 11-14 years 43% 11-14 years | 66.50% 53.3% | 3 months | Tiredness (39.0%) Headache (23.1%) Shortness of breath (23.4%) | N/A |

*(Continued)*

**Table 1.** (Continued)

| No | Study | Country income status | SARS-COV-2 confirmation | Study design | N | Female % | Age of participants | Persistent symptoms % | Duration of follow up | Three symptoms most reported % | Inpatient/ Outpatient/ Both |
|----|-------|----------------------|------------------------|--------------|---|----------|---------------------|----------------------|----------------------|-------------------------------|------------------------------|
| 38. | Sterky et al [69] | HIC | PCR (+) | PC | 55 | 42% | Range: 0–18 years | 22% | Min 4 months (median 219 days, range 123– 324 days) | Fatigue (66.7%) Myalgia (33.3%) Headache (33.3%) | Inpatient |
| 39. | Trapani et al [70] | HIC | PCR (+) | PC | 629 | 48.1% | Range: 0–16 years | 24.3% | >8–36 weeks | Fatigue (7%) Neurological (6.8%) Respiratory disorders (6%) | Outpatient |
| 40. | Zavala et al [71] | HIC | PCR (+) | PC Case Control | 859 472 387 | 52% 46% | Range: 2–16 years Median: 10 years Age-matched | 6.7% 4.2% | At least > 1 month | Anxiety (7%) Difficulty sleeping (7%) Tiredness (7%) | N/A |

HIC: high-income-country; LMIC: low-middle-income country; PCR: polymerase chain reaction; N/A: not available; IQR: interquartile range; Ab: antibody; PC: prospective cohort; RC: retrospective cohort; CS; cross sectional; NR; not reported.

61% among studies in only non-hospitalized patients [37,58] Most studies did not report the severity of symptoms.

All studies assessed outcomes (post COVID-19 conditions) for at least more than four weeks or a month. The longest follow-up duration was 13 months [53]. The most frequently used definition of post COVID-19 condition in children and adolescents was the persistence of symptoms for more than four weeks or a month following diagnosis of acute COVID-19 (used by 14 studies) [32,35,36,40,41,43,44,46,52,56,62,63,67,71]. Other cut-offs after diagnosis included two months (5 studies) [45,51,53,57,70], three months (11 studies) [33,34,37,42,47–49,54,64,65,68], four months [69] or five months (1 study each) [58] and six months or longer (6 studies) [39,50,55,59–61]. six months or longer (used by 6 studies) [39,50,55,59–61]. Two studies did not specify [38,66]. Among the studies, twenty-four studies follow WHO's criteria for symptom duration of "at least 2 months" but failed to specify the onset of symptoms. Therefore, it is difficult to identify which studies reported as consistent with WHO's definition of post COVID-19 condition.

## Persistent and common symptoms

Overall, the prevalence of persistent symptoms among all publications varied widely from 1.8% in 3,468 school children who had tested positive in the community in the UK and USA [57] to 70% in 378 Latvian children and adolescents followed after discharge from hospital [62] (Table 1). Similarly, the prevalence ranged from 1.8% to 66.5% in studies that reported symptom persistence of two months or more – i.e., as per WHO definition [57,58].

Among studies that included solely hospitalized patients, the prevalence of post COVID-19 condition ranged from 15% to 45% [34,41], whereas studies of only non-hospitalized patients reported a prevalence range of 13% to 61% [39,60]. Most studies did not provide detailed information on the severity of persistent symptoms.

The number of reporting publications for each persistent symptom type is shown in Fig 2. The most reported symptoms were fatigue (28/40, 70%), headache (15/40, 37.5%), and respiratory symptoms (14/40, 35%). Similar results were found from the 24 studies that reported consistent with WHO definition: fatigue [17/24, 70.8%], respiratory symptoms [10/24, 41.7%], and headache [9/24, 37.5%]. Within those studies, the prevalence of each symptom ranged widely from 1.1%–100% for fatigue [46,47]. 0.7%–55% for respiratory symptoms, [40,46] and

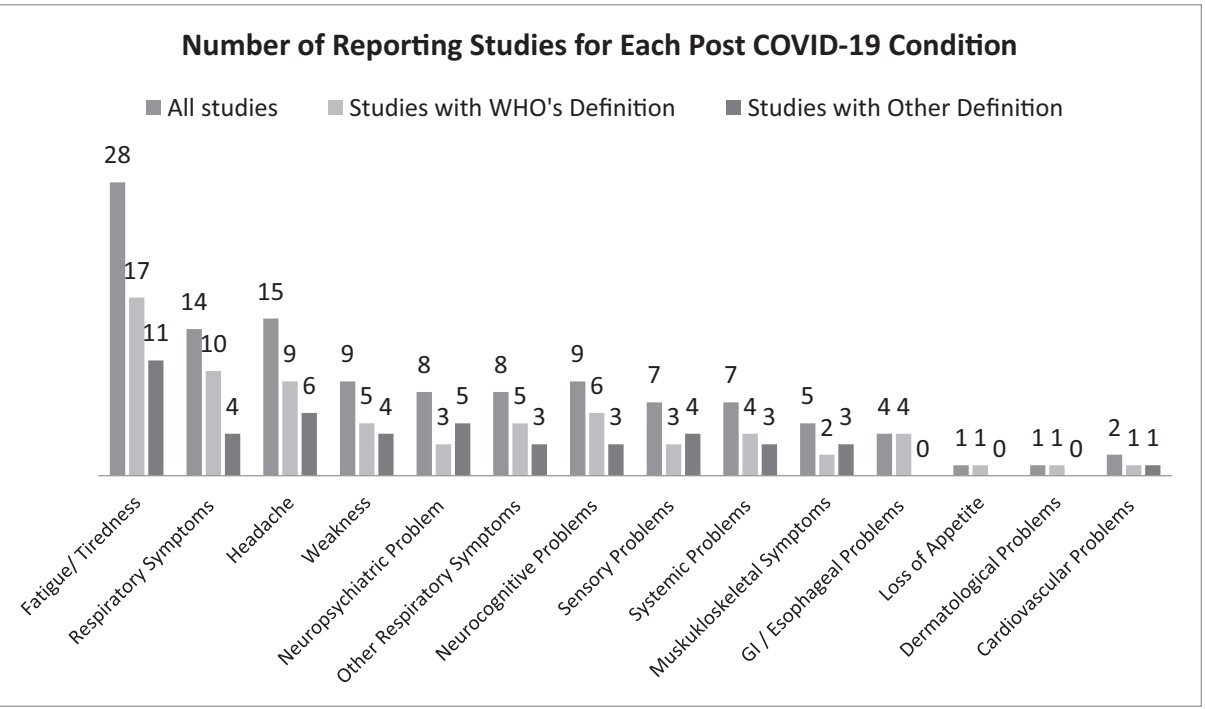

**Fig 2. The numbers of publications in which a specific symptom was among the three most commonly reported persistent symptoms.**

6–80% for headache [55,57]. The remaining 16 studies which do not meet WHO's definition of post COVID-19 condition found fatigue (11/16, 68.8%), headache (6/16, 37.5%), and neuropsychiatric problems (5/16, 31.3%) to be the most prevalent persistent symptoms. On the other side, the least reported symptoms in all studies include loss of appetite (1/40, 2.5%) and dermatological problems (1/40, 2.5%) [51,59].

## Gross national income

Publication on post COVID-19 condition included children or adolescents (<18 years) from 27 different countries but only one study was from a LMIC [34]. Five studies were from UMICs, [46,47,58,59,65], one study included populations from both UMICs and HICs [48], and the remaining 33 only included populations from HICs (Fig 3, S3 Table). In HIC studies, the prevalence of post COVID-19 symptoms ranged from 1.8% to 70% with fatigue (23/33 studies, 69.7%) and headache (reported in 13/33, 39.4%) being the most commonly reported symptoms. Within the five studies undertaken in an UMICs, the prevalence of post COVID-19 symptoms ranged from 20% to 37%, [46,59] with fatigue (reported in 4/5, 80%) and headache (reported in 2/5, 40%) as the most common symptoms. The single LMIC study conducted in Iran [34] and only included 58 participants who were all aged over five years; a post COVID symptoms were common - prevalence of 45% - and fatigue, shortness of breath and exercise intolerance were the commonest reported symptoms.

## SARS-COV-2 variants of concern

The distribution of variants across all included studies is detailed in Table 2. The predominating variant was not identifiable in 14 (35%) studies through electronic search because data of acute

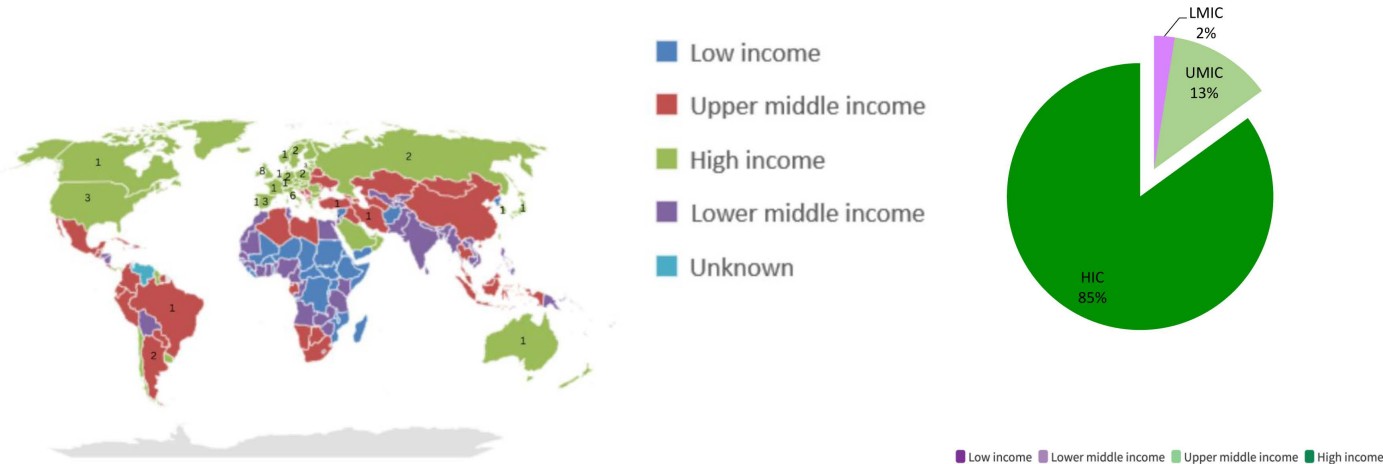

**Fig 3. Map and distribution of countries involved in post COVID-19 condition publications in children based on GNI by World Bank.**

COVID were derived within the first and second wave of the pandemic. In the remaining 26 studies, the Alpha variant was predominant in eleven studies [35,36,38,40,43,52,60,62,64,68,71], the Omicron variant in four studies [33,53,54,63], the Delta variant in four studies [32,47,51,70], the Gamma variant in one study [57], and other variants in six studies [42,48,58,59,61,65]. There is no beta variant-predominated study included in this review (Fig 4).

The range of prevalence of reported post COVID-19 condition was 4.1%–66.7% in the studies in which the variant of concern was not identifiable, [45,56] 6.7%–70% in the Alpha variant predominated studies, [62,71] 23%–61.9% in the Delta predominated studies, [45,49] 17–34.6% in the Omicron predominated studies, [33,54] and 3.7%–34% in the Other-variant predominated studies [61,65] Fatigue was the commonest symptom reported in all variant subgroups (9/11 [81.8%] in Alpha, 4/4 [100%] in Delta, 2/4 [50%] in Omicron, 6/6 [100%] in Other-variant predominated studies, and 7/14 [50%] in unidentifiable-variant-predominated studies).

## Other findings

Post COVID-19 conditions were more prevalent in older age groups [34,40,48,58] Most pediatric COVID-19 survivors reported one persistent symptom, followed by two with smaller numbers reporting three or more. Most symptoms at follow-up were reported as mild and tolerable by participants, though definition of mild and tolerable varied. Severe and disabling problems were reported in a few studies which were done in specialist medical centers that had referral bias for more. severe illness [34,35,42]. The severity of acute COVID-19 was associated with long-COVID-19 in a number of studies [34,37,48,58]. The prevalence of post COVID-19 symptoms tended to decline over time [43,58,59] and most post COVID-19 symptoms had resolved within a year [34,40,43].

## Quality of the study

Quality assessment based on adherence of STROBE recommendation for observational studies was presented in S2 Table. The mean score of all study was 35.0 (moderate quality). We categorized 10 (25%) studies as low quality (score ≤ 29) [33,35,41,42,54,61,64,66,67,69], 17 (42.5%) as moderate (score 30-39) [34,36–40,43–47,52,53,55,58,63,71] and 13 (32.5%) as high (score ≥ 40).

**Table 2. SARS-COV-2 variants of concern.**

| No | Study author | Study period | Density of circulating SARS-COV-2 Variants of Concern at Study Period | | | | | | Predominant variant |
|---|---|---|---|---|---|---|---|---|---|
| | | | Alpha | Beta | Delta | Gamma | Omicron | Other | |
| 1. | Adler et al | December 2021 to January 2022 | N/A | N/A | █ | N/A | █ | N/A | Delta |
| 2. | Ahn et al | May 2022 to July 2022 | N/A | N/A | N/A | N/A | █ | N/A | Omicron |
| 3. | Asadi-Pooya et al | February 2020 to November 2020 | N/A (First & second wave of pandemic) | | | | | | N/A |
| 4. | Ashkenazi-Hoffnung, et al | November 2020 to April 2021 | █ | █ | | N/A | N/A | █ | Alpha |
| 5. | Baptista de lima et al | March 2020 to September 2021 | █ | N/A | █ | █ | N/A | N/A | Alpha |
| 6. | Bergia et al. | March 2020 to December 2020 | N/A (First & second wave of pandemic) | | | | | | N/A |
| 7. | Bloise, et al | March 2020 to March 2021 | █ | █ | █ | █ | N/A | █ | Alpha |
| 8. | Blomberg et al | 28 February to 4 April 2020 | N/A (First wave of pandemic) | | | | | | N/A |
| 9. | Borch, et al | January 2020 to March 2021 | █ | █ | | N/A | | N/A | Alpha |
| 10. | Bossley et al | March 2020 to January 2021 | N/A (First wave of pandemic) | | | | | | N/A |
| 11. | Brackel, et al | December 2020 to February 2021 | █ | █ | █ | | N/A | █ | Other |
| 12. | Buonsenso, et al (a) | April 2020 to April 2021 | █ | | | | N/A | █ | Alpha |
| 13. | Buonsenso, et al (b) | UK: January 2020 to January 2021 | N/A (First wave of pandemic) | | | | | | N/A |
| | | US: January 2020 to January 2021 | N/A (First wave of pandemic) | | | | | | |
| 14. | Buonsenso, et al (c) | March to October 2020 | N/A (First wave of pandemic) | | | | | | N/A |
| 15. | Erol et al | March 2020 to February 2021 | N/A (First wave of pandemic) | | | | | | N/A |
| 16. | Fink, et al | April 2020 to August 2021 | █ | N/A | █ | █ | N/A | | Delta |
| 17. | Funk, et al | US: March 2020 to January 2021 | █ | █ | | | N/A | █ | Other |
| | | Costa Rica: March 2020 to January 2021 | N/A (First & second wave of pandemic) | | | | | | |
| | | Canada: March 2020 to January 2021 | █ | █ | N/A | N/A | N/A | █ | |
| | | Spain: March 2020 to January 2021 | █ | N/A | █ | N/A | N/A | █ | |
| 18. | Gonzales et al | December 2020 – May 2021 | N/A (First wave of pandemic) | | | | | | N/A |
| 19. | Haddad, et al | January 2020 to May 2020 | N/A (First wave of pandemic) | | | | | | N/A |
| 20. | Kikkenborg Berg, et al | Jan 2020 to July 2021 | █ | █ | █ | | N/A | | Delta |
| 21. | Knoke, et al | August 2020 to March 2021 | █ | █ | █ | | N/A | █ | Alpha |
| 22. | Kompaniyets, et al | March 2020 to January 2022 | N/A | N/A | █ | N/A | █ | █ | Omicron |
| 23. | Kuczborska et al | November 2020 and January 2022 | N/A | N/A | █ | N/A | █ | | Omicron |
| 24. | Matteudi et al | 27 February to 15 May 2020 | N/A (First wave of pandemic) | | | | | | N/A |
| 25. | Miller et al | June 2020 to May 2021 | N/A (First wave of pandemic) | | | | | | N/A |
| 26. | Molteni, et al | March 2020 to February 2021 | █ | █ | █ | █ | N/A | | Gamma |
| 27. | Osmanov, et al | April 2020 to August 2020 | N/A | N/A | N/A | N/A | N/A | █ | Other |
| 28. | Pazukhina, et al | April 2020 to August 2020 | N/A | N/A | N/A | N/A | N/A | █ | Other |
| 29. | Pereira et al | September 2020 to March 2021 | █ | | N/A | | N/A | N/A | Alpha |
| 30. | Radtke, et al | October to November 2020 | █ | N/A | N/A | N/A | N/A | █ | Other |
| 31. | Roge, et al | July 2020 to April 2021 | █ | N/A | N/A | N/A | N/A | | Alpha |
| 32. | Sakurada et al | Februari 2021 to October 2022 | █ | N/A | █ | N/A | █ | N/A | Omicron |
| 33. | Say, et al | March 2020 to March 2021 | █ | █ | █ | | N/A | | Alpha |
| 34. | Seery et al | June 2020 to June 2021 | N/A | N/A | N/A | █ | N/A | █ | Other |
| 35. | Smane et al (1) | July 2020 to July 2020 | N/A (First wave of pandemic) | | | | | | N/A |
| 36. | Smane, et al (2) | March 2020 to December 2020. | N/A (First wave of pandemic) | | | | | | N/A |
| 37. | Stephenson, et al | January 2021 to March 2021 | █ | █ | █ | | N/A | | Alpha |
| 38. | Sterky, et al | March 2020 to August 2020 | N/A (First wave of pandemic) | | | | | | N/A |
| 39. | Trapani et al. | June 2021 to August 2021 | █ | N/A | █ | | N/A | N/A | Delta |
| 40. | Zavala et al. | January 2021 | █ | █ | █ | | N/A | | Alpha |
| | **Predominant of SARS-COV-2 Variants** | | █ | █ | █ | █ | █ | N/A | |
| | | | > 50% | >30-50% | >20-30% | 10-20% | < 10% | | |

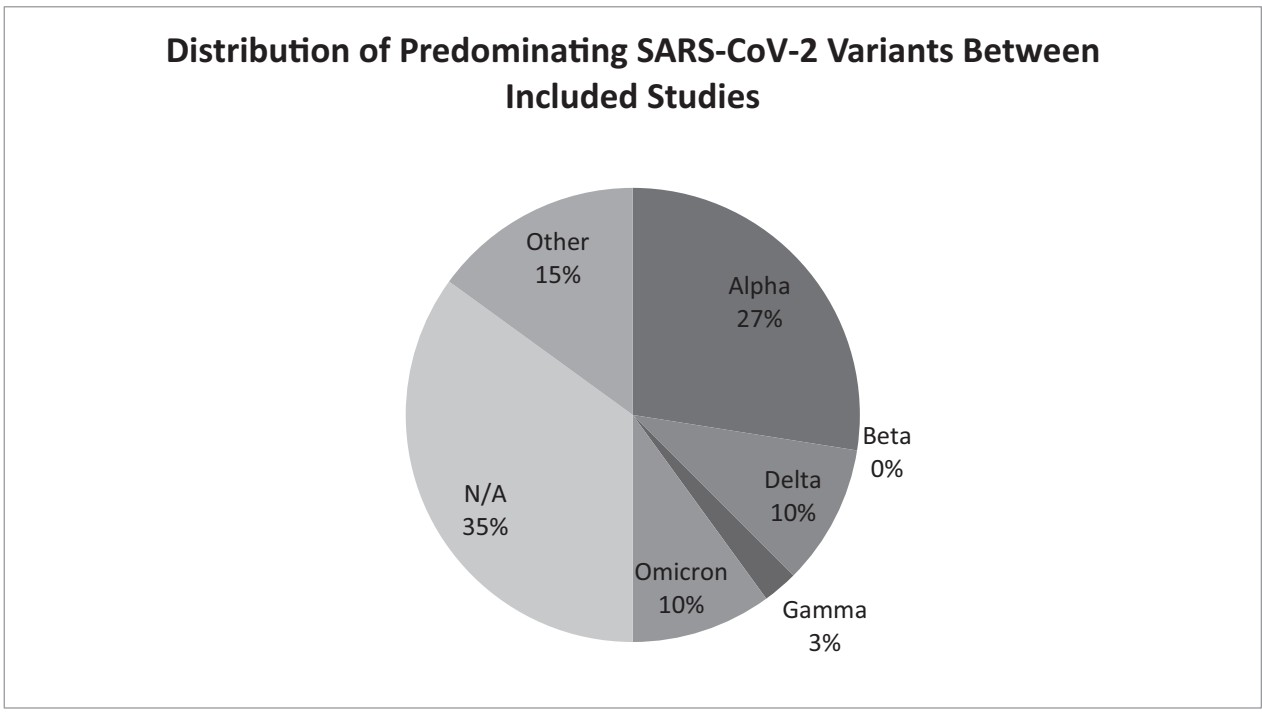

**Fig 4.  Distribution of predominating SARS-CoV2 variants between included studies.**

## Discussion

This systematic review provides a timely update on reported persistent symptoms in children and adolescents following infection with SARS-CoV-2. As SARS-CoV-2 testing for diagnosis and surveillance has diminished since 2022, new data are likely to become infrequent. Since the onset of the global pandemic in 2020, many studies have reported post-COVID symptoms in large numbers of children and adolescents from multiple countries. Our review includes five additional studies published since the most recent previous systematic review, but none from LMICs [15]. Though many studies were undertaken prior to the publication of the WHO consensus definition [18], most used similar criteria, especially as the consensus definition was informed by review of the same published studies. While there are major challenges for interpretation of data limited to symptom reporting, a standardized diagnosis definition is potentially helpful to enable meta-analyses, make comparisons with control populations and to measure impact of interventions.

Fatigue, headache and respiratory symptoms are consistently the most common persistent symptoms reported but there are major limitations relating to the interpretation of the findings. A reliance on symptom reporting alone, often with the lack of an appropriately matched control group, makes it challenging to define the true impact of COVID-19 infection on long-term health or recovery in children and adolescents. The symptoms may not be specific to post COVID-19 condition but could be influenced by numerous psychological effects of the pandemic or other pre-existing conditions of the subjects. The wide range of reported prevalence (1.8%–66.5%) of post COVID-19 condition likely reflects the heterogeneity between studies including study design (e.g., timing and method of follow-up assessment) and study participants such as variations in the severity of acute COVID. Fatigue is also the most prevalent post COVID-19 condition reported in adults along with joint pain, anosmia and headache [8,9]. In those studies that did include adults as well as children from the same

population, a significantly higher prevalence of persistent symptoms was reported in adults [39,43]. A more in-depth analysis comparing children to adults may be useful.

To date, the pathophysiology of post COVID-19 condition in children and adolescents is unknown [72]. A recent systematic review of six studies involving 678 adult COVID-19 survivors suggested a potential link between viral persistence and post COVID-19 conditions. In two months following acute infection, SARS-CoV-2 RNA was detected in 5% to 59% of patients with post COVID-19 condition, depending on the sample. However, the generalizability of the findings is constrained by the lack of a control group of individuals without post COVID-19 symptoms [73].

The systemic inflammatory response observed in acute COVID-19, characterized by elevated levels of cytokines, such as IL-2, IL-10, IL-6, IL-8, C-Reactive Protein (CRP), and Tumor Necrosis Factor (TNF)-alpha, triggers acute phase reactions and, if sustained, may lead to end-organ damage [74,75]. For instance, in lung tissue, the persistent elevation of proinflammatory cytokines is linked to a subset of pathological fibroblasts, which may contribute to the development of pulmonary fibrosis [76,77].

The severity of acute disease is indicated by the number of presenting symptoms, the need for hospitalization, the length of hospital stays, and the need for ICU admission [34,37,48,67]. It is hypothesized that severe COVID-19 triggers a more robust immune response with the resultant cytokine storm leading to more organ damage or that more aggressive and invasive treatment needed is more frequently associated with iatrogenic harm [78]. Increased prevalence of post COVID-19 condition in those with severe acute cases compared to a milder case has been also noted in adults [58]. Mitigating acute disease severity with antiviral treatments such as Remdesivir or Nirmaltrevir was not associated with reduced risk of developing post COVID-19 conditions [79].

Another mechanism of the dysregulated immune response is the development of an autoimmune response. Autoimmune diseases such as Guillain-Barré syndrome, Miller-Fisher syndrome, idiopathic thrombocytopenic purpura, Kawasaki-like disease, and MIS-C have been associated with COVID-19. However, a recent systematic review of five studies concluded that the clinical significance of persistent autoantibodies related to post COVID-19 conditions in adults appears limited and requires future research [80].

Different and more specific mechanisms have been proposed under each organ system in the adult population, and similarities between both groups have already been suggested [81]. Persistent fatigue, one of the most common complaints, arises from multifactorial causes, including brain hypometabolism, brain glymphatic drainage dysfunction, and brain toxin accumulation [82,83]. SARS-CoV-2 typically enters the central nervous system via hematogenous spread, promoting neuroinflammation and resulting in neuronal damage and neurologic symptoms [84]. Sustained neuroinflammation, whether through autoimmune reactions or activation of microglia, could account for neurocognitive and mental health disorders. Regarding the persistent taste and smell disorders, histologic assessments show persistent inflammation and-/or SARS-CoV-2 identification in the cells of taste buds and olfactory neuroepithelium [85,86].

Our review did not identify any additional studies from LMICs, highlighting the scarcity of published data and the uncertainty regarding the long-term health impacts of COVID-19 in these populations. Even before the COVID-19 pandemic, children and adolescents in LMICs had already faced a significant burden of severe disease and death due to infectious diseases [87]. Epidemiological and clinical risk factors, such as high exposure to indoor and outdoor pollution, as well as comorbidities associated with respiratory infections—including malnutrition, HIV infection, and tuberculosis [25,26]—may influence the clinical presentation and outcomes of respiratory infection, including those caused by SARS-CoV-2, in these vulnerable populations [88,89].

It is not known whether the prevalence or nature of post COVID-19 condition in children and adolescents varies according to the SARS-CoV-2 variant. Indeed, we observed a decline in the prevalence of post COVID-19 condition from studies dominated by the Alpha-variant to those dominated by the Omicron variant, a trend also reported in adults. This decline may be attributed to the availability of vaccination during the Omicron surge, which could have resulted in a milder disease course and fewer persistent symptoms [90]. A systematic review of six studies suggested that COVID-19 vaccination before acute infection was associated with reduced risks of long-COVID in adults, with two doses offering greater than one [91]. However, the COVID-19 vaccination status of children and adolescents was not provided in the included studies.

Analysis of post COVID-19 condition by sex and age in this review was limited. Univariate analysis by Asadi-Pooya et al. indicated a female-to-male prevalence ratio of 1.36 for post COVID-19 condition ($p < 0.441$) [34]. Kikkenborg et al. reported a higher proportion of female participants with symptoms persisting beyond two months after initial diagnosis: 71.7% vs 48.4% (OR 2.70; 95% CI 2.40–3.03; $p < 0.0001$) [51]. Previous meta-analysis has also supported female sex as a risk factor for developing post COVID-19 conditions in adults (OR 1.48, 95% CI 1.17 to 1.86). Further investigation is needed to elucidate the mechanism by which females are at increased risk of post COVID-19 condition, beyond the biological and sociocultural differences between sexes [91]. Moreover, the higher representation of the adolescent age group in the included studies may reflect either a greater susceptibility of adolescents to acute and post COVID-19 condition or the possibility that young children with acute COVID-19 were less likely to seek care or be identified, given the high prevalence of acute respiratory illnesses from other pathogens. Alternatively, this representation might also reflect the studies' reliance on symptom reporting [37]. Several studies primarily used self-reported questionnaire data, which increases the risk of recall bias.

## Supporting information

**S1 File. Search strategy.**
(DOCX)

**S2 Table. Quality of included studies.**
(DOCX)

**S3 Table. Additional study details.**
(DOCX)

**S4 Table. Abstract PRISMA checklist.**
(DOCX)

**S5 Table. Full Text PRISMA checklist.**
(DOCX)

**S6 Table. Details of excluded studies.**
(DOCX)

**S7 Table. Data extraction details.**
(DOCX)

## Acknowledgments

We would like to thank Poh Chua, medical librarian who helped us in literature search. This systematic review's formal protocol is part of the doctoral degree research proposal and has not been formally published. This review has also not been registered.

## Author contributions

**Conceptualization:** Nina Dwi Putri, Julie Bines, Stephen M. Graham.

**Data curation:** Nina Dwi Putri.

**Formal analysis:** Nina Dwi Putri.

**Funding acquisition:** Nina Dwi Putri, Dwiana Ocviyanti.

**Investigation:** Nina Dwi Putri, Julie Bines, Stephen M. Graham.

**Methodology:** Nina Dwi Putri, Julie Bines, Stephen M. Graham.

**Project administration:** Nina Dwi Putri.

**Software:** Nina Dwi Putri.

**Supervision:** Sri Rezeki Hadinegoro, Julie Bines, Stephen M. Graham.

**Validation:** Nina Dwi Putri, Julie Bines, Stephen M. Graham.

**Visualization:** Nina Dwi Putri.

**Writing – original draft:** Nina Dwi Putri, Stephen M. Graham.

**Writing – review & editing:** Nina Dwi Putri, Ida Safitri Laksanawati, Dominicus Husada, Nastiti Kaswandani, Ari Prayitno, Rina Triasih, Irma Sri Hidayati, Retno Asih, Robby Nurhariansyah, Fabiola Cathleen, Dwiana Ocviyanti, Sri Rezeki Hadinegoro, Dan Pelicci, Julie Bines, Stephen M. Graham.

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
