## [Decision Letter · Decision Letter 0]

13 Aug 2024

PONE-D-24-06176A systematic review of post COVID-19 condition in children and adolescents: Gap in evidence from low and -middle-income countries and the impact of SARS-COV-2 variantsPLOS ONE

Dear Dr. Putri,

Thank you for submitting your manuscript to PLOS ONE. After careful consideration, we feel that it has merit but does not fully meet PLOS ONE’s publication criteria as it currently stands. Therefore, we invite you to submit a revised version of the manuscript that addresses the points raised during the review process.

Authors have not provided sufficient literature review in the discussion. I encourage to incorporate more evidence, particularly with the newly updated ones.

We look forward to receiving your revised manuscript.

Kind regards,

Muhammad Iqhrammullah, Ph.D

Academic Editor

PLOS ONE

“Indonesia Endowment Fund/LPDP for publication support.”

“We would like to thank Poh Chua, medical librarian who helped us in literature search. The authors of this paper gratefully acknowledge Indonesia Endowment Fund/Lembaga Pengelola Dana Penelitian (LPDP) for PhD scholarship funding awarded to NDP.”

“Indonesia Endowment Fund/LPDP for publication support.”

5. We note that Figure 3 in your submission contain [map/satellite] images which may be copyrighted. All PLOS content is published under the Creative Commons Attribution License (CC BY 4.0), which means that the manuscript, images, and Supporting Information files will be freely available online, and any third party is permitted to access, download, copy, distribute, and use these materials in any way, even commercially, with proper attribution. For these reasons, we cannot publish previously copyrighted maps or satellite images created using proprietary data, such as Google software (Google Maps, Street View, and Earth). For more information, see our copyright guidelines: http://journals.plos.org/plosone/s/licenses-and-copyright.

1. You may seek permission from the original copyright holder of Figure 3 to publish the content specifically under the CC BY 4.0 license. 

Reviewers' comments:

Reviewer's Responses to Questions

**Comments to the Author**

1. Is the manuscript technically sound, and do the data support the conclusions?

Reviewer #1: Partly

Reviewer #2: Yes

2. Has the statistical analysis been performed appropriately and rigorously? 

Reviewer #1: No

Reviewer #2: Yes

3. Have the authors made all data underlying the findings in their manuscript fully available?

Reviewer #1: Yes

Reviewer #2: Yes

4. Is the manuscript presented in an intelligible fashion and written in standard English?

Reviewer #1: Yes

Reviewer #2: Yes

5. Review Comments to the Author

Reviewer #1: A systematic review of post-COVID-19 symptoms in children and adolescents, focusing on low- and middle-income countries (LMICs), analyzed 40 studies with 825,849 participants. Most studies involved adolescents, with limited data on young children. Only one study from an LMIC was identified, involving 58 participants. The studies varied in their definitions and durations of post-COVID-19 conditions, complicating comparisons. Prevalence ranged from 1.8% to 70%, with notable heterogeneity and reliance on self-reported symptoms. Fatigue, headache, and respiratory symptoms were the most common. The review highlighted the need for more data from LMICs and young children.

Although this study provided interesting insights, I recommend further revisions to refine the manuscript:

Recent studies have shown the possible persistence of long COVID for 2 years in the general population (DOI: 10.1016/j.jinf.2023.12.004). However, the meta-analysis data reported in this study is only limited to the adult population. It would be important to emphasize the need for a more in-depth analysis looking at the prevalence of long COVID symptoms in the pediatric age group and how these symptoms compare with the adult population.

Kindly include a supplementary section detailing the search strategy and the outcome of the meshed search terms for every journal database used.

Please include a PICO statement as well as the PRISMA checklist.

Kindly include the rationale for why a meta-analysis was not performed in this study. You can include this statement in the methods section.

Did the studies included in the systematic review undergo quality assessment, such as using the Newcastle-Ottawa scale, QUIPS, etc.? If not, please include a quality check for all the studies to assess the risk of bias.

This systematic review lacks in-depth analysis. It is important to report the pooled demographics of the children and adolescents included in this review.

Were there control groups included in the study? For example, children or adolescents who are either healthy or those who succumbed to an acute infection but did not develop long COVID?

What were the limitations of the present study? What are the gaps that need to be addressed in long COVID research involving these age groups?

There are many theories contributing to the development of long COVID. I think it is important to highlight these studies, which include the possible persistence of SARS-CoV-2 RNA (DOI: 10.1515/cclm-2024-0036), excessive inflammatory response, and even the generation and persistence of autoantibodies (DOI: 10.3389/fimmu.2024.1428645).

It is also important to highlight the several factors that can influence the severity and risk of developing long COVID, including vaccination prior to an acute infection (DOI: 10.1016/j.eclinm.2022.101624), administration of antivirals during the acute phase of SARS-CoV-2 infection (DOI: 10.1007/s15010-023-02154-0), and demographics like sex and certain comorbidities (DOI: 10.3390/jcm11247314), as well as the infecting SARS-CoV-2 variant (DOI: 10.3390/v14122629).

Reviewer #2: The manuscript has shown a good work of a scientific article, however, I noticed that there is still room for improvement, which can be found in the file I attached in this section. Please revise the manuscript accordingly, thank you!

6. PLOS authors have the option to publish the peer review history of their article (what does this mean? ). If published, this will include your full peer review and any attached files.

**Do you want your identity to be public for this peer review?** For information about this choice, including consent withdrawal, please see our Privacy Policy .

Reviewer #1: No

Reviewer #2: No

---

## [Author Response · Author response to Decision Letter 1]

27 Oct 2024

Rebuttal letter

Dear Editors,

We thank the reviewers for their generous comments on the manuscript and have edited it to address the concerns. The correction details are presented below.

We believe that the manuscript is now suitable for publication in PLOS One.

Sincerely,

Nina Dwi Putri

Reviewer 1.

1. Recent studies have shown the possible persistence of long COVID for 2 years in the general population (DOI: 10.1016/j.jinf.2023.12.004). However, the meta-analysis data reported in this study is only limited to the adult population. It would be important to emphasize the need for a more in-depth analysis looking at the prevalence of long COVID symptoms in the pediatric age group and how these symptoms compare with the adult population.

Response: Thanks for this important point. Specific reference to the adult data has been added in the Introduction, Lines 72-74. Emphasis of the need for more in-depth analysis in children with comparison to adult data is added in Discussion, with reference added to findings in studies that did include and report on adults in same population (refs 39 and 43). Lines 376-380.

2. Kindly include a supplementary section detailing the search strategy and the outcome of the meshed search terms for every journal database used.

Response: Thank you. This has now been added to Supplementary 1

3. Please include a PICO statement as well as the PRISMA checklist.

Response: A PICO statement has not been included as the data reviewed are observational. The Population line 120-124 and Outcomes line 127-130 are clearly defined and included. However, there is no Intervention being studied and so no Comparison can be made such as between age groups because of study heterogeneity. The characteristics of control groups (by age and sex) whenever reported are listed in Table 1 for the studies that compared persistence of symptoms to COVID-confirmed cases.

PRISMA Checklists have been added, which are:

• PRISMA Checklist for abstract (Supplementary 4)

• PRISMA Checklist for manuscript (Supplementary 5)

4. Kindly include the rationale for why a meta-analysis was not performed in this study. You can include this statement in the methods section.

Response: It was not possible to conduct a meta-analysis for reasons outlined in Methods line 226-229 which reads “We did not conduct a meta-analysis due to the wide heterogeneity in study design (such as varying definitions of post COVID-19 condition, timing and method of follow-up assessments) and in study participants (including variations in reporting by age groups and the severity of acute COVID).”

5. Did the studies included in the systematic review undergo quality assessment, such as using the Newcastle-Ottawa scale, QUIPS, etc.? If not, please include a quality check for all the studies to assess the risk of bias.

Response: There is a specific section in manuscript Line 187-188 under the sub-heading of “Assessment of Risk of Bias”, with detailed table included as Supplementary 2.

6. This systematic review lacks in-depth analysis. It is important to report the pooled demographics of the children and adolescents included in this review.

Response: Within the acknowledged limitations of the variation in reporting of data by the included studies, we have strived to report pooled demographics based on the data we extracted for participant details including by age, sex, definition, symptoms, diagnostic method and severity. Results paragraph: lines 245-251

7. Were there control groups included in the study? For example, children or adolescents who are either healthy or those who succumbed to an acute infection but did not develop long COVID?

Response: Thank you. We included all studies (with or without control) and the data regarding the control population when available are included in Table 1 such as age, sex and proportion with persistent symptoms when available. The statement can be found in Line 175-176.

8. What were the limitations of the present study? What are the gaps that need to be addressed in long COVID research involving these age groups?

Response: The limitations of available data are referred to repeatedly in the manuscript – the main ones being study heterogeneity not allowing meta-analysis and lack of clearly defined controls in many studies. Limitation of the included studies are presented separately in Line 363-366, Line 367-371, Line 382-387, Line 418-422, line 429-432 while limitation of this review is shown in Line 414-422, Line 433-435, Line 445-446

9. There are many theories contributing to the development of long COVID. I think it is important to highlight these studies, which include the possible persistence of SARS-CoV-2 RNA (DOI: 10.1515/cclm-2024-0036), excessive inflammatory response, and even the generation and persistence of autoantibodies (DOI: 10.3389/fimmu.2024.1428645).

Response: Thank you. These have been added to the manuscript. Line 382-387, Line 390-393, Line 400-413

10. It is also important to highlight the several factors that can influence the severity and risk of developing long COVID, including vaccination prior to an acute infection (DOI: 10.1016/j.eclinm.2022.101624), administration of antivirals during the acute phase of SARS-CoV-2 infection (DOI: 10.1007/s15010-023-02154-0), and demographics like sex and certain comorbidities (DOI: 10.3390/jcm11247314), as well as the infecting SARS-CoV-2 variant (DOI: 10.3390/v14122629).

Response: Thanks for these suggestions. The following have been added -

• Vaccination and SARS-CoV-2 variant has been added to Line 428-431

• Antiviral treatment has been added to Line 394-396

• Sex has been added to Line 433-441

We did not document comorbidities of included studies’ participants therefore did not discuss the influence of comorbidities to post COVID-19 condition

Reviewer 2.

11. Overall, this manuscript has given a clear and relevant explanation regarding the impact of SARS-CoV-2 variants in children and adolescence from HICs, UMICs, and LMICs. The manuscript is scientifically sound and the experimental design is reproducible. The data is interpreted appropriately throughout the manuscript. However, I have highlighted a room for improvement, which can be found below. Please address them accordingly. Thank you!

Response: thanks for this positive review.

12. (page 10, line 38): What does it mean with “1.8 to 70%”? Please unify the usage of percentage symbol to each mentioned numerical value.

Response: The format has been changed to 1.8% to 70%. All other percentages have also been changed to this format.

13. (page 10, lines 40-42): The repetition of “when Alpha predominated” from this sentence should be removed.

Response: This has been done. The line has been changed to: “The range of prevalence of reported post COVID-19 condition was 4.1%–66.7% in the studies in which the variant of concern was not identifiable, [45, 56] 6.7%–70% in the Alpha variant predominated studies, [62, 71] 23%–61.9% in the Delta predominated studies, [45, 49] 17–34.6% in the Omicron predominated studies, [33, 54] and 3.7%–34% in the Other-variant predominated studies [61,65] Fatigue was the commonest symptom reported in all variant subgroups (9/11 [81.8%] in Alpha, 4/4 [100%] in Delta, 2/4 [50%] in Omicron, 6/6 [100%] in Other-variant predominated studies, and 7/14 [50%] in unidentifiable-variant-predominated studies).”

14. (page 11, lines 61-62): This sentence “A systematic review of post COVID-19 condition indicates . .” will be more significant if it’s also supported by another relevant study. Please check https://doi.org/10.52225/narra.v1i3.48

Response: The study data are now referred to as: “A meta-analysis of 22 studies found that prolonged fatigue, joint pain, anosmia, headache and myalgia were recorded in 21.2%, 15.4%, 9.7%, 8.9% and 5.6% of adult COVID-19 survivors, respectively”.

15. (page 12, line 101): What is the meaning of “tw, kf, dq, and hw”? Please explain.

Response: now spelt out in text.

16. Please make sure that the percentage symbol is used with similar format throughout the manuscript. I noticed that the authors inconsistently use this symbol.

Response: All percentage format has been changed to “xx% to xx%

17. Authors are suggested to proofread the manuscript after addressing all comments to avoid any typo, grammatical, and lingual mistakes and errors.

Response: thank you for your feedback

18. (page 37, figure 4): Please make sure to apply the correct writing of SARS-CoV-2.

Response: it has been changed to: “Fig 4. Distribution of Predominating SARS-CoV-2 Variants between Included Studies”

19. (page 24, lines 358-360): This sentence “There are currently no published data . . .” is such a bold claim because I found a literature that reported a case of co-infection between SARS-CoV-2 and Orientia tsutsugamushi in a young teen in Nepal. Please check Bastola A, Sah R, Rajbhandari SK, et al. SARS-CoV-2 and Orientia tsutsugamushico-infection in a young teen, Nepal: Significant burden in limited-resource countries in Asia?

Response: thank you – now omitted

Answer: we have followed the styling guideline

21. Thank you for stating the following financial disclosure:

“Indonesia Endowment Fund/LPDP for publication support.”

Response: the funder has no role, the statement has been added in the manuscript and in the cover letter

22. Thank you for stating the following in the Acknowledgments Section of your manuscript:

“We would like to thank Poh Chua, medical librarian who helped us in literature search. The authors of this paper gratefully acknowledge Indonesia Endowment Fund/Lembaga Pengelola Dana Penelitian (LPDP) for PhD scholarship funding awarded to NDP.”

“Indonesia Endowment Fund/LPDP for publication support.”

Response: Funding information has been removed from the acknowledgement section.

23. We note that your Data Availability Statement is currently as follows: [All relevant data are within the manuscript and its Supporting Information files.]

Response: minimal data set (4 Microsoft excel worksheets) have been added, namely:

• Minimal Data 1_Post COVID-19 Condition based on SARS-CoV-2 Variants (data for Figure 2).xlsx

• Minimal Data 2_Post COVID-19 Condition based on Gross National Income

• Minimal Data 3_Predominating SARS-CoV-2 Variants Between Included Studies (Data for Figure 4)

• Minimal Data 4_Risk of bias assessment of included studies (For Supplementary 2)

24. We note that Figure 3 in your submission contain [map/satellite] images which may be copyrighted. All PLOS content is published under the Creative Commons Attribution License (CC BY 4.0), which means that the manuscript, images, and Supporting Information files will be freely available online, and any third party is permitted to access, download, copy, distribute, and use these materials in any way, even commercially, with proper attribution. For these reasons, we cannot publish previously copyrighted maps or satellite images created using proprietary data, such as Google software (Google Maps, Street View, and Earth). For more information, see our copyright guidelines: http://journals.plos.org/plosone/s/licenses-and-copyright.

Remove: The map has been redesigned to remove any copyright

---

## [Decision Letter · Decision Letter 1]

2 Dec 2024

A systematic review of post COVID-19 condition in children and adolescents: Gap in evidence from low and -middle-income countries and the impact of SARS-COV-2 variants

PONE-D-24-06176R1

Dear Dr. Putri,

We’re pleased to inform you that your manuscript has been judged scientifically suitable for publication and will be formally accepted for publication once it meets all outstanding technical requirements.

Kind regards,

Muhammad Iqhrammullah, Ph.D

Academic Editor

PLOS ONE

Additional Editor Comments (optional):

Reviewers' comments:

Reviewer's Responses to Questions

**Comments to the Author**

1. If the authors have adequately addressed your comments raised in a previous round of review and you feel that this manuscript is now acceptable for publication, you may indicate that here to bypass the “Comments to the Author” section, enter your conflict of interest statement in the “Confidential to Editor” section, and submit your "Accept" recommendation.

Reviewer #1: All comments have been addressed

Reviewer #2: All comments have been addressed

2. Is the manuscript technically sound, and do the data support the conclusions?

Reviewer #1: Yes

Reviewer #2: Yes

3. Has the statistical analysis been performed appropriately and rigorously? 

Reviewer #1: Yes

Reviewer #2: Yes

4. Have the authors made all data underlying the findings in their manuscript fully available?

Reviewer #1: Yes

Reviewer #2: Yes

5. Is the manuscript presented in an intelligible fashion and written in standard English?

Reviewer #1: Yes

Reviewer #2: Yes

6. Review Comments to the Author

Reviewer #1: The manuscript has been extensively revised and is now acceptable for publication. Congratulations to the authors.

Reviewer #2: Thank you for addressing my concerns! Now the manuscript has improved a lot, especially in terms of its quality and readability. I would suggest it for publication.

7. PLOS authors have the option to publish the peer review history of their article (what does this mean? ). If published, this will include your full peer review and any attached files.

**Do you want your identity to be public for this peer review?** For information about this choice, including consent withdrawal, please see our Privacy Policy .

Reviewer #1: No

Reviewer #2: No

---

## [Editor Report · Acceptance letter]

PONE-D-24-06176R1

PLOS ONE

Dear Dr. Putri,

I'm pleased to inform you that your manuscript has been deemed suitable for publication in PLOS ONE. Congratulations! Your manuscript is now being handed over to our production team.

Kind regards,

on behalf of

Dr. Muhammad Iqhrammullah

Academic Editor

PLOS ONE